# A Change in Conservation Status of *Pachyphytum caesium* (Crassulaceae), a Threatened Species from Central Mexico Based on Genetic Studies

**DOI:** 10.3390/biology11030379

**Published:** 2022-02-27

**Authors:** Tania Martínez-León, Ricardo Clark-Tapia, Jorge E. Campos, Luz Isela Peinado-Guevara, Samuel Campista-León, Francisco Molina-Freaner, Nelly Pacheco-Cruz, Gabriel González-Adame, Juan José Von Thaden Ugalde, Cecilia Alfonso-Corrado

**Affiliations:** 1Instituto de Estudios Ambientales, Universidad de la Sierra Juárez, Avenida Universidad S/N, Ixtlán de Juárez, Oaxaca 68725, Mexico; tanleo.m@gmail.com (T.M.-L.); rclark@unsij.edu.mx (R.C.-T.); gaboadame@unsij.edu.mx (G.G.-A.); juan.vonthaden@unsij.edu.mx (J.J.V.T.U.); 2Unidad de Biotecnología y Prototipos, Facultad de Estudios Superiores de Iztacala, Universidad Nacional Autónoma de México, Avenida de los Barrios 1, Los Reyes Iztacala, Tlaneplantla 54090, Mexico; jcampos@unam.mx (J.E.C.); nelly.pacheco.cruz@gmail.com (N.P.-C.); 3Laboratorio de Microbiología y Biología Aplicada, Facultad de Biología, Universidad Autónoma de Sinaloa, Av. Universitarios, S/N, Ciudad Universitaria, Culiacán Rosales 80013, Mexico; luzipg@uas.edu.mx (L.I.P.-G.); samcl@uas.edu.mx (S.C.-L.); 4Departamento de Ecología de la Biodiversidad, Instituto de Ecología, Universidad Nacional Autónoma de México, Hermosillo 83250, Mexico; freaner@unam.mx

**Keywords:** biodiversity loss, cliff-dwelling, conservation, extinction, genetic drift, risk evaluation method, threatened plant

## Abstract

**Simple Summary:**

Species decline has accelerated in recent decades, with rare species often being the first to go extinct, mainly due to low population sizes. This study worked with *Pachyphytum cesium* (Crassulaceae), an endemic species from central Mexico threatened by human activities and climate change. To increase our knowledge of the species, we analyze the diversity and genetic structure of all known populations of *P. caesium* to change their current genetic status and propose conservation strategies. The results indicate that this species presents low to moderate levels of genetic diversity and incipiently structured populations due to low genetic flow. We found that genetic parameters are essential to the conservation strategies and status vulnerability reclassification. Based on our results, we propose reclassifying the conservation status of the species in danger of extinction; hence a series of conservation strategies are provided to prevent its disappearance.

**Abstract:**

Genetic studies in rare species are important to determine their patterns of genetic diversity among populations and implement conservation plans aimed to reduce extinction risks. *Pachyphytum caesium* is an endemic species with extremely small populations in central Mexico. This work analyzes the diversity and genetic structure of *Pachyphytum cesium* (Crassulaceae) to change the conservation status and propose conservation strategies. Twelve dominant ISSR markers were used to describe the genetic diversity of all known populations. Additionally, we carried out two simulations to explore how the loss of individuals or the local populations extinction affect the genetics parameters of the species. The average results indicated moderate levels of genetic diversity (number of alleles = 89.7 ± 3.9, expected heterozygosity = 0.212 ± 0.0, and percentage of polymorphic loci = 56.1 ± 3.9), parameters that decreased significantly when simulations were performed in *P. caesium*. Additionally, a genetic structure of the populations was found with low gene flow (Nm = 0.92). Genetic parameters are negatively affected by the loss of individuals and the local extinction of populations. Based on our results, we propose to reclassify the conservation status of the species in danger of extinction, hence a series of conservation strategies are provided to prevent its disappearance.

## 1. Introduction

Ecological and genetic studies are essential to implement conservation plans, especially in endangered, rare and endemic plant species [1,2,3], in order to face of the global crisis of biodiversity loss [4,5]. Genetic studies of rare plants allow us to describe patterns of genetic diversity, population structure, and effective population size [1,2,6]. Thus, these studies can significantly strengthen strategies for population management, determine critical populations for conservation, and reduce species’ risk of extinction [4,5,7]. Moreover, genetic studies are essential when the evaluation of extinction risk is difficult to assess from ecological and demographic data alone [7,8].

Currently, there are various methods of DNA-based markers for the genetic analysis of plant populations, the most common being amplified-fragment length polymorphism (AFLP), inter-simple sequence repeats (ISSRs), microsatellites (SSR), and single nucleotide polymorphisms (SNPs); each varies in polymorphism quantification, repeatability, resolution power and analysis cost [9,10,11]. Among these markers, ISSR is a multi-dominant locus marker that generally produces multiple DNA fragments throughout the genome of any species, presents Mendelian inheritance, reproducibility, is highly polymorphic and low cost [10,11,12]. Although a major drawback of ISSRs is its inability to detect heterozygotes, this marker is recognized as a valuable and affordable method for the evaluation of diversity and genetic structure of a wide range plants [13], including Crassulaceae species [14].

Although genetic studies are important toward the species conservation, these results are rarely included in the Red List of threatened species of the International Union for Conservation of Nature’s (IUCN) methodology [8,15,16,17]. Several authors have suggested the importance of incorporating genetic information in the red list criteria, since currently considered parameters are insufficient criteria of the ecological and evolutionary resilience of the species to global change, including land-use and climatic changes [16,17]. Presently, the Red List is focused on recent, current, or future threats related to sustainable habitat loss, population size, or exploitation levels [18].

Nowadays, habitat loss and climate change are two of the major causes of the current biodiversity crisis, as they cause not only cascading extinctions [1,2,3], but also local extinctions or a drastic reduction in the effective population size with effects on the evolutionary potential of a species [5,19,20]. In this context, the vulnerability to extinction is the primary concern in Plant Species with Extremely Small Populations (PSESP). PSESP species are characterized mainly by small population sizes, low genetic variation, restricted distribution, habitat loss, and high vulnerability to environmental changes that make them particularly vulnerable to extinction [21,22,23].

Habitat loss has been particularly worrying in central Mexico, due to land-use change, as more than 80% of its original vegetation has been transformed into agriculture and cattle raising [21,24]. This activity has been threatening an area with high species diversity [21,25], including species of *Pachyphytum*, an endemic genus of the Crassulaceae [26]. Knowledge of this genus is mainly limited to morphological descriptions [26,27,28,29] and a general description of the current distribution of their species [29]. Among these species, *Pachyphytum caesium* Kimnach & Moran is a cliff-dwelling species endemic to the dry tropical forest of Aguascalientes in central Mexico, and has been categorized as PSESP due to its highly specialized habitat, small and fragmented populations, and a high vulnerability to environmental changes [21]. Based on the risk of extinction evaluation method of wild species (MER in Spanish), Clark-Tapia et al. [21] classified the status of this species as threatened and recommended incorporating the species into the Mexican federal protection list.

The MER is a tool that is applied exclusively in Mexico, as part of the Mexican Official Standard (NOM-059-SEMARNAT-2010) [30]. This tool evaluates the status of a taxon in the under-country parameters, and according to Feria Arroyo et al. [31], the MER compared to the IUCN system provide similar results. However, one aspect that distinguishes MER is the consideration of the genetic factors in the intrinsic biological vulnerability section.

Because an appropriate estimation of extinction risk is one of the key components of any conservation plan, in this work we evaluated the diversity and genetic structure of the currently know populations of *Pachyphytum caesium* reported by Clark-Tapia et al. [21]. In addition, we assessed the effect of habitat loss on genetic diversity by simulating the loss of individuals in a population or local population extinction in the data set to evaluate its effect on genetic parameters in the MER and provide recommendations for in situ conservation.

## 2. Materials and Methods

### 2.1. Study Area

The study area is located in central Mexico, between 22°40′00″ N and 102°30′00″ W in the southwestern portion of the state of Aguascalientes (Figure 1). The only six known sites with *P. caesium* are located in this region [21], whose natural range is marginal (less than 20 ha) and fragmented in that territory. The study area has a subtropical climate, with rains in the summer (from July to November) with an average annual rainfall of 591.4 mm, and a temperature ranging from 15 to 29.5 °C. The altitudinal distribution at the study area ranges from 1600 to 2400 masl (meters above sea level) (Figure 1). The dry tropical forest is highly fragmented and covers some 178.94 km^2^, representing 3.18% of the state’s total surface [25].

### 2.2. Species Description

*Pachyphytum caesiu**m* (Crassulaceae) is a succulent perennial plant growing in the north-central region of Mexico [26,28]. A general plant description is provided in Figure 2. The species has a restricted distribution to rocky cliffs in tropical dry forest and its transition with oak forest [27,28,29] within the ecoregions of semiarid southern elevations and temperate areas of Mexico [32]. Population sizes vary between sites and can range from less than 150 individuals to more than 6000 individuals in size. The smallest populations are Barranca Tortugas, Presa Cebolletas, and Mesa Montoro, while Río Gil, Puente Cuates, and Presa Malpaso have the highest densities [21]. The area of each population varies from 1 to 5 ha, and they are remnants of vegetation due to land-use change. Although the average distance between sites is 5.99 ± 2.8 km, the populations are separated by natural barriers (hills or mountains), including areas with agriculture and livestock.

This species presents two flowering periods per year (from February to April and from September to November). The reproduction is sexual, and the observed pollinators are bees and hummingbirds. The fruit produces numerous small (<1 mm) brown seeds. However, their reproductive biology is unknown, and more studies are needed [21]. Additionally, *P. caesium* can be propagated through asexual reproduction, from the leaves or stems that are detached (Pers. Observations).

### 2.3. Biological Tissue Collection

The tissue collection of *Pachyphytum caesium* was carried out between 2009 and 2013 as new populations were found. We collected young leaves of 20 individuals per population, for a total of 120 individuals. Distance separation between collected individuals was 20 m to avoid kinship. To access each site and collect the tissue, we used a rock-climbing technique with a Petzl stop descender. Subsequently, we stored the collected biological material in a plastic bag. Once in the laboratory, samples were stored at −70 °C to preserve it until used.

### 2.4. Genetic Analysis

DNA was extracted following the Qiagen DNeasy Plant Kit protocol (Qiagen N.V., Hilden, Germany). For the evaluation of the quality and quantity of genetic material obtained from the 120 individuals, electrophoresis was performed in 1% agarose (Invitrogen, Fisher Scientific, Waltham, MA, USA) gels in TBE buffer 0.5X, and stained with Midori Green Direct (Nippon, Genetic Cop. Ltd., Düren, Germany) for an average of 40 min at 90 V. Subsequently, these gels were visualized with UV light in a transilluminator equipment Gel Doc XR + System (BIO-RAD, Hercules, CA, USA).

For this study, different primers of the ISSR dominant marker obtained from the database of the University of British Columbia were evaluated. Given financial limitations preventing the use of more robust markers, the threatened status and priority of studying *P. caesium*, we opted for ISSR, a dominant marker with reasonable reproducibility and polymorphism than other markers, as well as a robust pattern in the analysis of genetic diversity [9,10,11,14,33]. Identification of the 12 ISSR primers utilized in this study resulted from preliminary amplification of 30 *P. caesium* individuals; these were selected for their high-polymorphic states and repeatability (Appendix A).

PCR reactions were performed in 25 μL of reagent (Appendix A). The description of the PCR program employed is described in Appendix A. The amplified DNA segments obtained with this technique were separated using electrophoresis in 1.4% agarose (Invitrogen, Fisher Scientific, Hampton, NH, USA) gels, in TBE 0.5X. They ran at 100 V for two and a half hours. The gels were made using Midori Green Direct (Nippon, Genetic Cop. Ltd., Tokyo, Japan), and the band pattern was documented using a photo documentation system (Gel Doc x R + System) (BIO-RAD Hercules, CA, USA). The digital image files were transferred in SCN format and analyzed using Image Lab TM software v.5.2.1 (BIO-RAD Hercules, CA, USA). This software is able to detect the presence of clear and well-defined bands, as well as the fragment sizes in the photos of the gels and exported the data to an Excel spreadsheet, which was validated through a peer review procedure. The banding patterns obtained by Image Lab TM software were used to generate a binary data matrix. In the matrix, the presence of a band was considered as “1” and the absence of it as “0”. The matrix was used for the genetic analysis.

#### 2.4.1. Allelic and Genetic Diversity

The number of total alleles (Io), effective number of alleles per population (Eff_num), and private alleles per population (Ie) were determined as the total number of observed bands (It) using GenoDive v.3.0 [34]. In addition, the exact differentiation test was applied [35] to verify significant differences in allelic frequencies among populations.

Moreover, we calculated the following genetic diversity parameters: (1) percentage of polymorphic loci with 95% criterion (%P) and (2) expected heterozygosity (He) [6] using the Lynch and Milligan [36] correction, and (3) the Shannon diversity index (I). These indices were calculated using the GenAlEx v.6.501 program [37].

#### 2.4.2. Population Structure and Isolation by Distance

We calculated the population structure in six ways: (1) the level of genetic differentiation among populations was obtained with the estimator Gst [38], using the software GenoDive [34]. (2) The analysis of molecular variance with 10,000 permutations (AMOVA) and descriptor Fst of genetic differentiation [39] were assessed using the GenAlEx [37]. Since the database obtained with ISSR is binary, we select the binary option in the analysis. (3) The Minimum Spanning Network (MSN) determined the minimum genetic distance between individuals and populations based on the Hamming dissimilarity index [40]. The analysis was performed in R v.3.6.2 [41] using the adegenet [42] and Poppr [43] packages.

Additionally, we used (4) the principal component analysis (PCA) to analyze genetic variation and possible associations in the allele frequencies of the sampled populations using a covariance matrix [41]. (5) We performed a discriminant analysis of principal components (DAPC) to identify and describe genetically-related groups. Since DAPC requires prior groups to be defined, the clustering by k-means and the Bayesian information criterion were used to define the number of groups before the analysis. In addition, the gene flow was calculated according to Crow and Aoki [44]. Both analyzes were performed in R [41] with the adegenet package [42], and its visualization with the ggplot2 package [45].

Finally, (6) we used a Bayesian model-based clustering approach to identify the genetic structure in populations of *P. caesium*. It was performed using the software STRUCTURE v.2.3.4 [46]. We used the admixture model, with ten independent replicate runs per K value ranging from 1 to 10 to run the structure algorithm. Each run involved a burning period of 100,000 iterations and a post-burning simulation length of 500,000. The most probable number of clusters (K) was estimated using STRUCTURE HARVESTER [47], which implements the Evanno method [48] to determine the optimal K depending on the probability of the data for a given K and the ΔK.

To evaluate whether *P. caesium* shows a pattern of isolation by distance, a Mantel test with 10,000 permutations was performed with the APE package [49], using the association between the geographic distance matrix based on the logarithm of geographic distances (linear distance between populations) and genetic matrix (linearized Slatkin Fst differentiation) [50]. Subsequently, the Spearman correlation coefficient between these matrices was calculated. In addition, 95% confidence intervals were generated to evaluate the significance of the correlation (P). All analysis was carried out in R [41].

#### 2.4.3. Simulation to Obtain Genetic Diversity and Structure

In 2018, we returned to the previously collected populations and found a drastic decrease in abundance (≥50% in two sites), in addition to an extinct local population reported by Clark-Tapia et al. [21]. Given the lack of economic resources to carry out a new genetic analysis, it was decided to simulate the genetic effect caused by the extinction of two local populations that were reported in critical condition (Presa Cebolletas and Mesa Montoro). In addition, we simulated the effect of random loss of half of the individuals in each population from the original database. The extraction of individuals was carried out using random numbers. Subsequently, to parameterize the post-2018 scenario, we calculated the diversity and genetic structure variables for each simulation based on the methodology outlined above.

### 2.5. Conservation Status

Based on the method for evaluation of the risk of extinction of species in the Mexican legislation (MER, Appendix II of NOM-059-SEMARNAT-2010) [30], we estimated the conservation status of *Pachyphytum caesium*. This tool used four normalized criteria to assess risk categories of species: (A) breadth distribution; (B) habitat status; (C) intrinsic biological vulnerability (ecological and genetic); and (D) human impact. Numerical values of each criterion are assigned in ascending order of risk, which are subsequently added up. Each criterion is evaluated qualitatively, and the total score is calculated by adding the normalized results from the four. The higher the total score value, the higher the risk of extinction of the assessed species. According to the MER [30], species with a total score ≥ 2.0 points are considered as in danger of extinction (EE), as threatened (E) between points ≥ 1.70 and < 2.0, whereas a score ≤ 1.70 and ≥ 1.5 can grant special protection status (Pr).

We analyzed the MER for two years (2005 and 2020e) based on the ecological and genetic knowledge of the species. We used 2005 because we only had basic knowledge (breadth distribution and habitat status). In 2020e, we analyzed the MER criteria using only ecological knowledge [14] and ecological data with loss of individuals (2020e_50_) and local extinction (2020e_ex_). In addition, we integrated ecological and genetic data, considering average values (2020e + g) and two simulations, one with loss of individuals (2020e + g_50_) and the other with extinction (2020e + g_ex_). For comparison, we also applied parts of the IUCN Red List system [18]. This method contemplates five criteria: population size, geographic range, small population size and decline, very small or restricted population, and quantitative analysis. However, unlike MER, the method does not require all criteria to be completed to achieve an assessment [30].

## 3. Results

### 3.1. Allelic and Genetic Diversity

An average of allele number of 89.7 ± 3.9 was found for the populations studied (Appendix A). Five private alleles were detected (one and four in Puente Cuates and Cebolletas, respectively). The population of *Pachyphytum caesium* that presented the highest number of alleles was Presa Cebolletas (96), and the lowest was Presa Malpaso and Mesa Montoro (85 and 86, respectively). The total number of alleles did not change when performing the simulation with random extraction of 50% of the individuals or the local extinction of two populations (Presa Cebolletas and Mesa Montoro). However, the number of private alleles decreased significantly (χ^2^ = 15.23; *p* < 0.05) (Appendix A).

On the other hand, the highest levels of genetic diversity were found in Presa Cebolletas (He = 0.297; I = 0.398; %P = 73.27), and the lowest in Mesa Montoro (He = 0.187; I = 0.272; %P = 46.53), with a low population average (He = 0.208 ± 0.03, I = 0.307 ± 0.01 and %P = 56.1 ± 3.86). The expected heterozygosity was significantly more affected by the simulated extraction of individuals than by the local extinction of two populations (Appendix A). Furthermore, no significant differences were found in allelic frequencies between populations in the total analysis and the simulations.

### 3.2. Population Structure and Isolation by Distance

The analysis of molecular variance (AMOVA) showed a significant (*p* < 0.001) partitioning among the populations, of which 43% was attributed among populations and the rest 57% within populations. The coefficients of genetic differentiation among populations Gst and Fst were 0.471 and 0.443, indicating that more than 44% variation was presented among the populations, which is indicative of population structuring. Identical values of variation between populations (43%, *p* < 0.001) were obtained after the random extraction of individuals or the local extinction of populations.

Figure 3 shows the minimum spanning network for the species *Pachyphytum caesium*, where a well-defined structuring of the six populations was found. The populations whose genotypes had the greatest number of interconnections with genotypes from other populations were Mesa Montoro and Presa Malpaso, unlike Puente Gil, Presa Cebolletas, and Barranca Tortugas that were connected to just one population. On the other hand, Mesa Montoro showed the most significant differentiation of multilocus genotypes on a population level, compared to Río Gil, that also showed a greater closeness to Mesa Montoro (Figure 3a). In simulations using extraction of individuals (Figure 3b) and local population extinction (Figure 3c), we registered a change in the distance between individuals and populations. In these cases, a decrease in the number of interconnections between individuals from each population was observed.

The principal component analysis (PCA) explained 23.98% of the accumulated variance, where the first component explained 13.56% of the total variation, while the second explained 10.42%. The analysis generated four well-defined population groups, including the association between Puente Cuates, Presa Malpaso, and Río Gil, which is closely associated with Mesa Montoro (Figure 4a).

In the PCAs obtained with the simulations of the extraction of individuals (Figure 4b) and the local extinctions (Figure 4c), the total variance increased slightly (25.11 and 32.06%, respectively); the first component being the best explained in both cases. However, there was no change in the number and grouping between populations, except that the Presa Malpaso population separated itself as an independent group, with the local extinction of two populations.

Finally, discriminant analysis of principal components (DAPC) detected that the optimal number of principal components to generate the model was two. This result generated a model of two discriminant functions (horizontal and vertical) where a grouping pattern between populations like the PCA was obtained and there was a strong structuring between populations (Figure 5a).

With the simulations of individual extraction (Figure 5b), the grouping pattern was inverted in the discriminant functions of the DAPC. In addition, the populations of Presa Cebolletas and Puente Cuates are isolated in the first discriminant function on the horizontal axis (Figure 5b). In the DAPC with the two local populations extinction, two populations located in the same axis were separated, Barranca Tortugas stayed on the second discriminant function on the vertical axis, while Río Gil was transferred to the horizontal axis on the first function (Figure 5c). In general, cluster analysis showed that Presa Malpaso, Río Gil, and Puente Cuates were the closest populations, while Mesa Montoro was intermediate, and Presa Cebolletas and Barranca Tortugas were the most differentiated populations.

The maximum value of ΔK, mean(LnProb), and mean (similarity score) were detected at K = 2 by Evanno’s method using the total individuals and simulated 50% of populations, indicating that two groups could be distributed across all the *P. caesium* populations (Figure 6a,b). A detailed study of the clusters revealed that group-1 (blue) contained the populations of Presa Malpaso, Puente Cuates, Río Gil, and Mesa Montoro. In contrast, group-2 (orange) distinctly had the populations of Barranca Tortugas and Presa Cebolletas (see Figure 1). Under the two populations extinction scenario, the maximum value of ΔK, mean(LnProb), and mean(similarity score) were detected at K = 3. In this scenario, group-1 (blue) contained the populations of Puente Cuates and Río Gil, group-2 (purple) had distinctly to Barranca Tortugas, and group-3 was Presa Malpaso (Figure 6c).

Gene flow between the populations was Nm = 0.92 migrants per generation for all individuals or considering 50% of the individuals per population. However, Nm decreased dramatically to 0.633 when we simulated a local extinction of two of the six populations.

Mantel’s correlation analysis found no correlation between genetic and geographic distances considering the six populations (r = 0.357; *p* = 0.29; 999 permutations), or 50% of the individuals (r = 0.421; *p* = 0.19), and the local extinction of two populations (r = 0.557; *p* = 0.25).

### 3.3. Conservation Status

The MER showed four results according to the criteria and knowledge used in the method: (1) with basic knowledge of two criteria (A: breadth distribution and B: habitat status), the species was classified in special protection category (2005). (2) Adding the ecological knowledge about intrinsic biological vulnerability and human impact (criteria C and D, respectively), the status of the species was elevated to threatened (2020e). (3) After incorporing the genetic knowledge into intrinsic biological vulnerability (criteria A, B, C, and D), the status of the species was further elevated to in danger of extinction level (2020e + g). Although the inclusion of genetic factors in both simulations also generated a status of danger of extinction, the total score in vulnerability was higher (2020e + g_50_ and 2020e + g_ex_) (Figure 7).

On the contrary, the IUCN system showed three conservation statuses: (1) the status of vulnerable for years with basic knowledge (2005), ecological (2020e), and ecological-genetic (2020e + g). (2) The endangered status obtained in the simulations with a decrease in population size (2020e_50_ and 2020e + g_50_), and (3) the critically endangered status achieved with simulations of local extinction (2020e_ex_ and 2020e + g_ex_).

## 4. Discussion

### 4.1. Allelic and Genetic Diversity

The results of this study show that the ISSR method is suitable for exploratory genetic studies of plant species because it was able to detect a genetic diversity comparable to other DNA-based markers [9,10,11,12]. The knowledge of genetic parameters in plant species with extremely small population size provides important information supporting recommendations for their conservation, e.g., [19,23], and lays the groundwork for proposing the reclassification of the risk status of a particular species, e.g., [51]. In addition, genetic studies on endemic species have allowed a better understanding of their response to environmental changes and decreased their extinction risk [1,2,3,7]. In the particular case of *Pachyphytum caesium*, its genetic diversity values were low to moderate but similar or greater than that reported in PSESP species with an analogous molecular marker [9,10,11,19,23,52] and other members of the Crassulaceae family [14,53,54,55].

Some studies suggest that endemic species or species cataloged as PSESP present lower levels of population genetic diversity than species with larger populations and continuous distribution, a result attributed to the species’ life history [19,23,53,56]. In this context, several studies suggest that low genetic variation in plants is not always associated with low sexual reproductive success, e.g., [57,58]; however, the absence of demographic stability could be a factor for the loss of diversity due to gene drift, as suggested by Charlesworth et al. [59] and Chung et al. [54]. In clonal species as *P. caesium*, with low or no sexual recruitment in their populations [19], its current diversity would be residual and maintained by clonal reproduction as suggested by Clark-Tapia et al. [19]. In this case, clonal propagation can buffer the extinction of the species and moderate the loss of alleles due to genetic drift in small populations [19,60,61], but it can negatively influence the genetic structure [19]. However, it is an aspect that should be studied in the future with a more powerful molecular marker such as single nucleotide polymorphisms (SNPs).

Currently, Presa Cebolletas is the population with the greatest genetic diversity (heterozygosity, total alleles and private alleles); however, it shows an absence of sexual recruitment despite producing seeds. For this reason, we suggest that the small populations’ size of *P. caesium* and the long history of land-use change in the region are the major causes of genetic diversity differences among their populations [21]. The history of disturbance has differed between populations [21] and is recent in Presa Cebolletas. Based on the high values of genetic diversity of Cebolletas (see Appendix A), we suggest that the Presa Cebolletas region served as a geographic refugium in the past, like some postglacial genetic refuge of temperate plant species [62]. Its higher allelic diversity and rare allele presences support that Presa Cebolletas worked as the original population [63]. Furthermore, recolonization processes may be the cause of differences in diversity between populations. The founder effect can generate a decrease in genetic variation and changes in allele frequencies or loss of private alleles [19,64]. This scenario is possible for *P. caesium* populations, in synergy with the loss of alleles due to disturbance, an aspect that should be studied in the future.

Land-use change or anthropogenic activities have been documented to affect the distribution and size of a population and species’ genetic diversity [2,23,54,65]. For example, Vogler and Reich [65] found that rappelling as an anthropic short-time factor that decreases genetic diversity of *Draba aizoides*, a cliff-dwelling species of the Northern Franconian Jura and Swabian Alps in Germany due to the decrease in the population number of adults and seedlings. Our results suggest a similar scenario of affectation at the genetic level, considering the simulation results with loss of individuals and local extinction. It is known that the populations of *P. caesium* have decreased due to changes in land use and sport climbing [21]; however, we need to analyze in the future the effect of these activities at the genetic level.

### 4.2. Population Structure and Isolation by Distance

An incipient level of among-population genetic differentiation was revealed in *Pachyphytum caesium*, like other endangered or endemic species, such as *Megacodon stylophorus* (Gentianaceae) [66] and *Rhodiola alsia* (Crassulaceae) [14]. The genetic structure obtained in the study suggests a low dispersal of seeds and pollen, despite the absence of genetic isolation by distance. Geographic isolation due to natural barriers or suitable habitat loss may significantly affect the genetic structure of plant populations [23,54]. *P. caesium* populations have a fragmented distribution due to land-use changes, with a discontinuous distribution associated with cliffs in the remnants of the tropical dry forest in the state of Aguascalientes [21]. The lack of genetic studies in the genus *Pachyphytum* makes its comparison difficult. However, it is possible with other endemic plants or PSESP that inhabit cliffs, including members of the Crassulaceae family. In these species, it is common to find a high genetic differentiation between populations, e.g., [52,54,61,66].

In *Pachyphytum caesium*, the absence, or low gene flow, as a factor that could homogeneize allelic frequencies [6,67], together with the absence of isolation by distance, suggests effects due to genetic drift. This evolutionary force is reported in other plant species with small, fragmented populations and low gene flow [68,69]. Genetic drift increases divergence between populations due to gene flow restriction and generates a loss of genetic diversity caused by allele fixation in a few generations [6,67]. This loss of connectivity between *P. caesium* populations, added to the allelic loss due to the decline in their populations and local extinctions, increases the genetic drift effect and allows the action of other evolutionary forces as inbreeding. If this is so, new molecular studies (SNPs) are also required to know aspects of inbreeding, genetic landscape, and locate local and functional genotypes to provide greater certainty in the conservation of the species.

Even though there is a discrepancy in the conformation of groups between the Minimum spanning network, PCA and DAPC analyses showed between 3 and 4 groups, concerning STRUCTURE that obtained two groups, or three in the simulation without two populations. At landscape level, the cliffs from Río Gil, Presa Malpaso, and Puente Cuates have greater connectivity between them as they are within the same micro-basin and are more likely to exchange migrants. Therefore, they have a greater association and overlap extensively in the analyses. In contrast, the three remaining populations have been isolated in recent decades with less probability of genetic exchange.

Cluster analysis and STRUCTURE clustering showed a clear geographic pattern suggesting that, in the past, *P. caesium* populations were fragmented, leading to isolation and genetic drift phenomena as reported in other endemic species [70]. A possible explanation for the difference between association analysis and STRUCTURE could be the different resolution between analyses. For example, although STRUCTURE is amongst the most frequent genetic method, its results can be influenced by the effective size, bottleneck severity, and the number of loci; according to Lombaert et al. [71], further study is needed to clarify the differences.

### 4.3. Genetics Studies and Conservation Implications

This study highlights the importance of using genetic data in the risk analysis of a species, as suggested by Garner et al. [8], Victorino et al. [16], and Lanes et al. [17]. The genetic results and simulations included in the MER [26] suggest that the species is in danger of extinction and not threatened, as reported by Clark-Tapia et al. [21] using only ecological data. This reassessment contrasts with the IUCN system [18], where it remained in the same category (VU), considering basic ecological, demographic, and genetic data. The inclusion of genetic data in the IUCN Red List methodology could increase the risk of *P. caesium*, a result found in other studies, which also emphasizes the need to consider genetic studies.

Like the MER, the risk is increased in the IUCN by modifying the population size and distribution area; two key criteria used by these methods assign threat status [18,30]. Although both criteria are directly linked to heterozygosity [16,72], other critical genetic and evolutionary processes such as inbreeding, genetic drift, or loss of heterozygosity are overlooked [16,17,73]. Thus, some authors suggest that if a species is re-assessed with genetic considerations, their status risk can change [16,17]. However, it is essential to note that in the IUCN system, the change from the lower risk categories to the higher risk categories can be changed annually if the species knowledge is increased [18]. However, this is not the case in the list of the Mexican legislation, which is modified every three years [30]. Hence, it is of vital importance to include genetic factors when determining a realistic picture of the conservation status of the species, e.g., [8,16,73].

In PSESP, as in *P. caesium*, precautionary conservation measures should be appropriate because the risk of extinction is higher for small populations [22,23], which are more susceptible to genetic drift due to inbreeding [74] and the local alleles loss [75]. That is why, although ISSR markers provide an alternative approach for cost-effective detection of DNA polymorphisms, essential for the status risk evaluation in this study. Future studies should evaluate the use of SNPs to study other genetic parameters, such as inbreeding, effective population size, or specific SNPs “outliers”, subject to selection, giving information of the species capacity to adapt to future environmental changes.

The immediate conservation strategy is to insert the species on the IUCN red list and the list of species at national risk, in addition to the studies of the species’ vegetative propagation and reproductive biology which have already begun. The establishment of rappelling routes and the prohibition of this activity in some areas where the species is found is a pending activity. Given the possibility that *P. caesium* is not subject to federal protection in the short term, its vulnerability to adapt to future environmental changes increases, mainly due to a decrease in private alleles, e.g., [76], and reduction of genetic variation and the lack of conservation programs based on genetic information. Therefore, another central strategy that is being implemented is collecting seeds and tissues for their conservation in a germplasm bank, and genetic information might guide such collections [77], which will allow future restoration programs.

## 5. Conclusions

Our study found that *Pachyphytum caesium* exhibit moderate to low levels of genetic diversity, a strong genetic structure of their populations, and low gene flow. Genetic drift is an evolutionary force that negatively affects the genetic population structure and depends on the demographic stability of the species. The simulations of extraction of individuals and the local extinction of populations negatively affect genetic diversity, which should be considered for population conservation. According to ecological and genetic data, populations of *P. caesium* are at risk of extinction following the MER. We highlight the importance of using genetic data in MER, which is why we suggest its use in the IUCN Red List methodology. Finally, genetic data can support future endangered species conservation planning.

## Figures and Tables

**Figure 1 biology-11-00379-f001:**
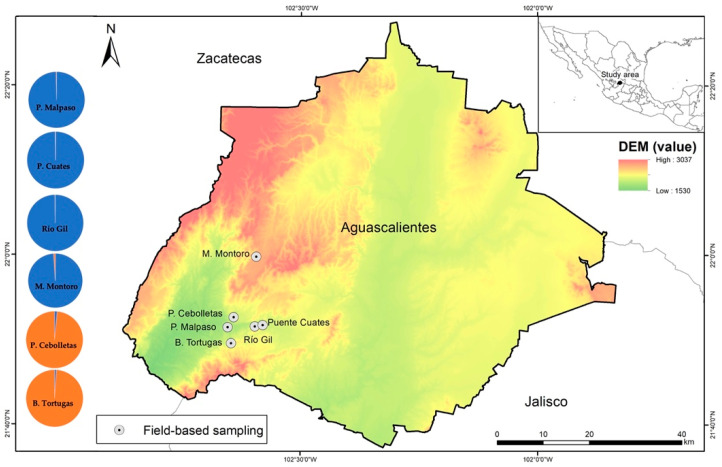
Geographic distribution of the six studied populations of *Pachyphytum caesium*. DEM: digital elevation model with altitudes ranging from 1530 to 3037 masl. Pie charts indicate the mean proportion of the membership of individuals at each site for K = 2 genetic groups inferred by Bayesian clustering implemented in STRUCTURE from current data. P.: Presa Cebolletas and Presa Malpaso; B.: Barranca Tortugas, and M.: Mesa Montoro.

**Figure 2 biology-11-00379-f002:**
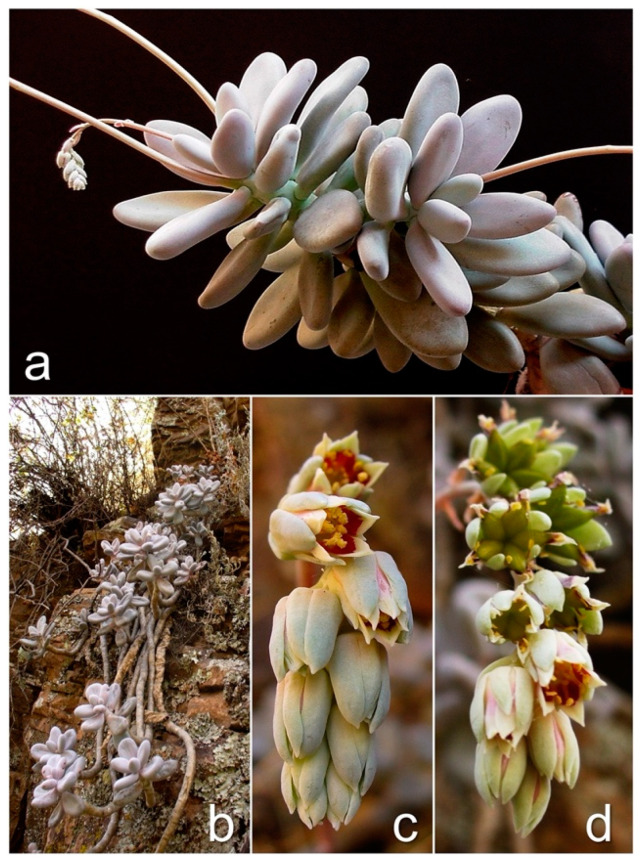
*Pachyphytum caesium* Kimnach & Moran. (**a**) Leaves obovate-oblong with subobtuse apex, grayish to grayish-purple, 3–7 cm, forming lax terminal rosettes; axillary inflorescences of long-stalked cincinnus, simple, 6–20 flowers. (**b**) Rupicolous plant, stems commonly branched basally, erect in young plants but elongated and pendulous with age, reaching 30–40 cm. (**c**) Flowers short pedicellate, tubulars, perianth almost equal in length, corolla cream to cream-green with a reddish spot centrally; nectaries yellowish [26,27]. (**d**) Fruits of polyfollicles with many oblong brownish seeds.

**Figure 3 biology-11-00379-f003:**
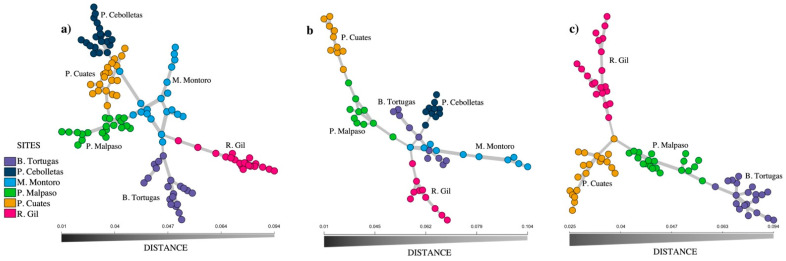
Minimum spanning network of the six populations of *Pachyphytum caesium*, without simulation (**a**); with simulation of the extraction of individuals (**b**); and the local extinction of two populations (**c**).

**Figure 4 biology-11-00379-f004:**
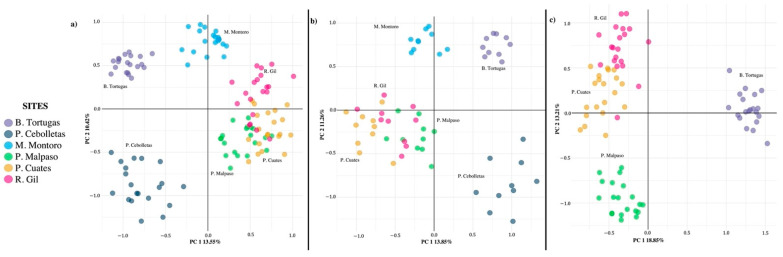
Principal component analysis of *Pachyphytum caesium* populations without simulation (**a**); with simulation of the extraction of individuals (**b**); and the local extinction of two populations (**c**).

**Figure 5 biology-11-00379-f005:**
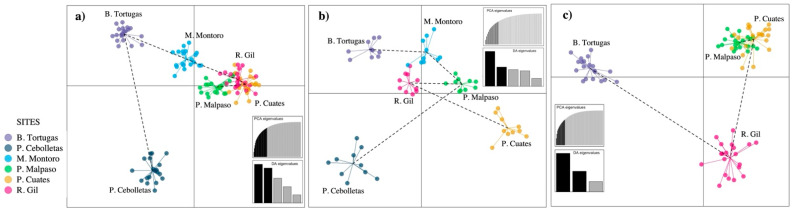
Discriminant analysis of principal components (DAPC) of *Pachyphytum caesium* without simulation (**a**); with simulation of the extraction of individuals (**b**); and the local extinction of two populations (**c**).

**Figure 6 biology-11-00379-f006:**
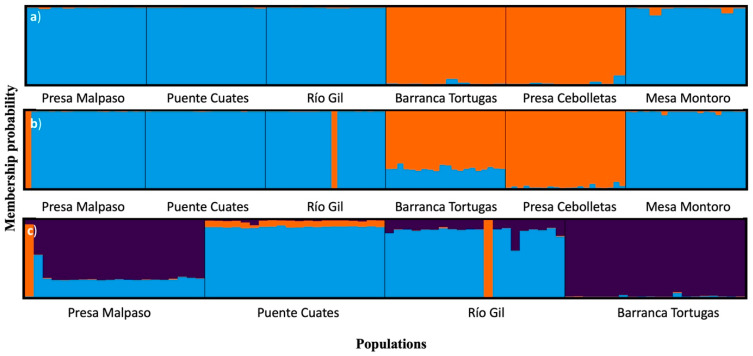
Genetic relationships among the populations of *Pachyphytum caesium* estimated using the STRUCTURE program based on ISSR data: (**a**) without simulation (120 individuals, see Figure 1); (**b**) with simulation of the extraction of individuals (60 individuals); and (**c**) the local extinction of two populations. The model with K = 2 and K = 3 showed the highest ΔK value.

**Figure 7 biology-11-00379-f007:**
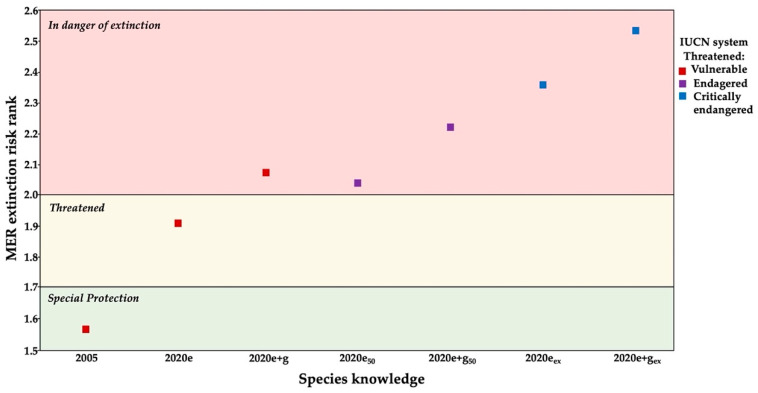
Comparison of the Mexican national system (MER) and IUCN Red List in *Pachyphytum caesium* among years based on ecological and genetic knowledge. Ecological data, average (2020e), with loss of 50% of individuals (2020e_50_) and local extinction of two populations (2020e_ex_); ecological and genetic data, average (2020e + g), with loss of 50% of individuals (2020e + g_50_) and local extinction of two populations (2020e + g_ex_).

## Data Availability

The data presented in this study are available in the main text and within the Appendix A section. The data can be provided by the authors if necessary.

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
