# Peer review of "A Change in Conservation Status of Pachyphytum caesium (Crassulaceae), a Threatened Species from Central Mexico Based on Genetic Studies"

_biology, 2022, doi:10.3390/biology11030379_

Round 1
Reviewer 1 Report
This study presents a genetic diversity assessment of six populations of Pachyphytum caesium, a cliff dwelling Crassulaceae that is endemic to the dry tropical forest of central Mexico. The species was first described in 1993 and much remains unknown about its reproductive biology and genetic diversity. Previous studies have shown that populations of P. caesium are very fragmented. Some populations are reported to have disappeared since 1995 (Clark-Tapia, R., et al., 2021, 10.3390/d13090421), likely because of land-use changes that have reduced the area of dry tropical forest in central Mexico and because of sport activities that disrupt cliff ecosystems (e.g., rock climbing).
The study presents valuable knowledge on the genetic diversity of a rare taxon that could soon be extinct in the wild. In my opinion, although the molecular marker system used to assess genetic diversity (i.e., ISSR) is outdated, the authors’ experimental design and findings are sound. The low genetic diversity values detected are remarkable, especially since ISSR is known to be highly polymorphic. It should also be noted that this study comes from a budget-limited setting —the authors make this explicit in the text (lines 128-132)—. Studies on rare, remote, and/or potentially endangered taxa from the developing world are likely to present such budget limitations and use somewhat outdated methods; however, the information they provide may not become available to the scientific community otherwise.
My main concern is that some important parts of the text are unclear because of grammar and syntax issues. I suggest that the authors make a thorough style correction. Here I detail more in this regard and list other concerns:
- In the abstract, the distinction between the full analysis and the simulations could be stated with more clarity. When I first read it, I had the impression that the full study was based on the simulations.
- Lines 139-150: except for the final reaction volume, the PCR protocol is presented in an unusual way. Instead of volumes, I suggest presenting the total amount of template DNA in nanograms and the final concentrations of the reagents of the PCR reaction (e.g., micromolarity and enzyme units). I suggest also avoiding double parentheses.
- Lines 147-150: the thermocycler program is confusing. What does “37°C of each primer” mean?
- Lines 212-219: how large was the “drastic” decrease in abundance? Is that reduction more or less than that of the scenario where half the individuals were randomly extracted?
- Lines 249-250: when the authors mention the average number of alleles per locus it is not clear why there are two different values. Is the second value derived from the six populations?
- Lines 267-271: the authors claim that the AMOVA results show that there is a “high population structuring” because 43% of the variance occurs among populations, while 57% occurs within populations. In my opinion, these results are indicative of the opposite: that AMOVA did not detect population structure. Could this be because the R. Gil, P. Cuates, and P. Malpaso populations form a cluster?
- Lines 303-314: more detail is needed when describing the DAPC results. Were the groups determined a priori or discovered de novo? The authors mention that clustering by k-means (line 190) was used to define the number of groups, but figure 4 shows all the sampling sites as clusters. The figure gives the impression that groups were determined a priori.
- In figure 5, the y-axis is wrongly labelled as “Delta K”. I think it should be the “membership probability”. In addition, “ΔK” is sometimes written as such (e.g., line 321), as “Delta K” (e.g., in the y-axis of figure 5), and as “DK” (e.g., line 338). I suggest choosing a single nomenclature.
- Only K = 2 is stated in the figure caption, while the text says that a K = 3 was used for the extinction scenario (line 329). Also, the “c)” inside the third panel is not legible.
- I suggest revising for clarity the whole section 3.3 “Status of conservation”. For this, I suggest using the nomenclature in figure 6 (e.g., 2005, 2020e, etc.) and a clearer sentence structure. Furthermore, the listing of results and criteria using letters (e.g., “a)” and “A:”) was confusing to me.
- Regarding figure 6: in my opinion, a dot plot is not the best representation for this kind of data (a categorical variable vs a semiquantitative score). I suggest replacing figure 6 with a table, which would make the message clearer.
- Lines 440-444: in my view, more detail is needed in describing the clustering patterns of the populations. Why do samples from R. Gil, P. Malpaso, and P. Cuates overlap extensively in the DAPC?
- Lines 493-499: the conclusions omit what, in my view, is one of the main findings of the study: that according to ecological and genetic data, the studied P. caesium populations are at risk of extinction according to the MER system. Furthermore, what implications do the simulations have in the real world (for example, in relation to the local extinction of populations that has been observed since 1995)?
Minor comments:
- Lines 112-114: to improve clarity, I suggest breaking down long statements into separate sentences.
- Line 123: is it “TBE medium” or “TBE buffer”?
- The original citation spells “GenAlEx” not “GenAlex” (Peakall, R., Smouse, P. E., 2012, https://doi.org/10.1093/bioinformatics/bts460).
- Software versions are sometimes written with a space (e.g., “GenoDive v. 3.0”, line 174) and sometimes without the space (e.g., “GenAlex v.6.501”, line 169). The “v” is sometimes capitalized. I suggest choosing a consistent pattern.
- Lines 188-191: the R citations in the text are unusual. The “adegenet” citation is missing.
- Lines 207-208: the authors state “the Spearman correlation coefficient r was calculated for each pair of matrices and log distance”. What kind of matrices and log distances?
- Lines 234-243: to improve clarity, I suggest using the nomenclature used in figure 6 (e.g., 2005, 2020e, 2020e+g, etc.).
- Line 317: there is an extra “)”.
- The acronym “SNPs” is sometimes presented as “SNP’s” (line 392) and “SNPs” (line 437). It should be “SNPs”.
- Lines 393-395: the sentence “Although the differences...” is unclear.
- Lines 410-413: this study did not assess the effect of land use change and rock climbing. I suggest clearly distinguishing the findings of this study from those of previous ones.
- Line 416: the word “high” should not be capitalized.
- Lines 440-444: I found it difficult to understand this paragraph. What does “grouping by the similarity between analyses” mean?
- Line 441: it should be “DAPC”.
- Lines: 476-479: I suggest using a more impersonal language when stating the outlook of the study.
Author Response
Response to Reviewer 1 Comments
General comments
Point 1. In the abstract, the distinction between the full analysis and the simulations could be stated with more clarity. When I first read it, I had the impression that the full study was based on the simulations.
Response 1. We reviewed and clarified the redaction (P1, Lines 28-30).
Point 2. Lines 139-150: except for the final reaction volume, the PCR protocol is presented in an unusual way. Instead of volumes, I suggest presenting the total amount of template DNA in nanograms and the final concentrations of the reagents of the PCR reaction (e.g., micromolarity and enzyme units). I suggest also avoiding double parentheses.
Response 2. We reviewed and modified the redaction in this section (P5, Lines 169-173). Also, we add a Table S1, Table S2 and Table S3 to supplementary material in order to clarify and make simple this information.
Point 3. Lines 147-150: the thermocycler program is confusing. What does “37°C of each primer” mean?
Response 3. We reviewed and clarified the redaction according to the reviewer's comment (P4, Lines 153-155). Also,we add Table S4 to supplementary material to address this Point
Point 4. Lines 212-219: how large was the “drastic” decrease in abundance? Is that reduction more or less than that of the scenario where half the individuals were randomly extracted?
Response 4. We have corrected the redaction accordingly (Line 231, Page 6)
Point 5. Lines 249-250: when the authors mention the average number of alleles per locus it is not clear why there are two different values. Is the second value derived from the six populations?
Response 5. We have made correction according to the Reviewer’s comments (Line 269, Page 7).
Point 6. Lines 267-271: the authors claim that the AMOVA results show that there is a “high population structuring” because 43% of the variance occurs among populations, while 57% occurs within populations. In my opinion, these results are indicative of the opposite: that AMOVA did not detect population structure. Could this be because the R. Gil, P. Cuates, and P. Malpaso populations form a cluster?
Response 6. We have made correction according to the Reviewer’s comments to clarify this Point. Also, considering the reviewer's observation, we consider that there is indeed no "high population structuring", however, given the information provided by the gene flow and the field observations where we have seen that there is both sexual and asexual reproduction, we can say that “there is structuring, although incipient” (P7, Lines 286-292). The possible reason for this grouping is mentioned in the discussion section (P12, Lines 471-478).
Point 7. Lines 303-314: more detail is needed when describing the DAPC results. Were the groups determined a priori or discovered de novo? The authors mention that clustering by k-means (line 190) was used to define the number of groups, but figure 4 shows all the sampling sites as clusters. The figure gives the impression that groups were determined a priori.
Response 7. We have corrected according to the Reviewer’s comments to clarify the text in Material and Methods section (Page 5, Lines 209-213).
Point 8. In figure 5, the y-axis is wrongly labelled as “Delta K”. I think it should be the “membership probability”. In addition, “ΔK” is sometimes written as such (e.g., line 321), as “Delta K” (e.g., in the y-axis of figure 5), and as “DK” (e.g., line 338). I suggest choosing a single nomenclature.
Only K = 2 is stated in the figure caption, while the text says that a K = 3 was used for the extinction scenario (line 329). Also, the “c)” inside the third panel is not legible.
Response 8. We adapted the redaction in this section to use a single nomenclature (ΔK). Also, Figure 5 was modifiedaccording to the Reviewer’s comments.
Point 9. I suggest revising for clarity the whole section 3.3 “Status of conservation”. For this, I suggest using the nomenclature in figure 6 (e.g., 2005, 2020e, etc.) and a clearer sentence structure. Furthermore, the listing of results and criteria using letters (e.g., “a)” and “A:”) was confusing to me.
Response 9. We reviewed and modified the redaction in this section according to Materials and Methods instructions, and clarified the redaction (Page 10, Lines 370-379).
Point #10. Regarding figure 6: in my opinion, a dot plot is not the best representation for this kind of data (a categorical variable vs a semiquantitative score). I suggest replacing figure 6 with a table, which would make the message clearer.
Response 10. We consider that a visual representation, such as the figure, allows a better interpretation of changes in scale and conservation status between years according to our knowledge. This type of interpretive figure has also being used by Arroyo et al., [31]. However, for the sake of comparison, below is what the table would look like:
Table 1. Comparison of the Mexican national system (MER) and IUCN Red List in Pachyphytum caesium among years based on ecological and genetic knowledge. Ecological data, average (2020e), with loss of 50% of individuals (2020e50) and local extinction of two populations (2020eex); Ecological and genetic data, average (2020e+g), with loss of 50% of individuals (2020e+g50) and local extinction of two populations (2020e+gex).
|
MER |
IUC |
|
|
2005 |
Special protection |
Vulnerable |
|
2020e |
Threatened |
Vulnerable |
|
2020e50 |
In danger of extinction |
Vulnerable |
|
2020eex |
In danger of extinction |
Endagered |
|
2020e+g |
In danger of extinction |
Endagered |
|
2020e+g50 |
In danger of extinction |
Critically endagered |
|
2020e+gex |
In danger of extinction |
Critically endagered |
Point 11. Lines 440-444: in my view, more detail is needed in describing the clustering patterns of the populations. Why do samples from R. Gil, P. Malpaso, and P. Cuates overlap extensively in the DAPC?
Response 11. We reviewed the redaction according to the reviewer’s comments to clarify the text (Page 12, Line 471-478).
Point 12. Lines 493-499: the conclusions omit what, in my view, is one of the main findings of the study: that according to ecological and genetic data, the studied P. caesium populations are at risk of extinction according to the MER system. Furthermore, what implications do the simulations have in the real world (for example, in relation to the local extinction of populations that has been observed since 1995)?
Response 12. We agree with the reviewer and modified the redaction in the Conclusion section (Page 13, Lines 540-547).
Point 13. Lines 112-114: to improve clarity, I suggest breaking down long statements into separate sentences.
Response 13. We reviewed the redaction according to the reviewer’s comments to clarify the text (Page 4, Line 149-155)
Point 14. Line 123: is it “TBE medium” or “TBE buffer”?
Response 14: We made correction, Page 4, Lines 160.
Point 15. The original citation spells “GenAlEx” not “GenAlex” (Peakall, R., Smouse, P. E., 2012, https://doi.org/10.1093/bioinformatics/bts460).
Response 15. We changed “GenAlex” by “GenAlEx”
Point 16. Software versions are sometimes written with a space (e.g., “GenoDive v. 3.0”, line 174) and sometimes without the space (e.g., “GenAlex v.6.501”, line 169). The “v” is sometimes capitalized. I suggest choosing a consistent pattern.
Response 16: We choose the option “v.” to describe the software versions, which was modified in the text.
Point 17. Lines 188-191: the R citations in the text are unusual. The “adegenet” citation is missing.
Response 17: We reviewed and corrected the R citation and packages in the section method (P6, Lines 205 and 213).
Point 18. Lines 207-208: the authors state “the Spearman correlation coefficient r was calculated for each pair of matrices and log distance”. What kind of matrices and log distances?
Response 18: We have made correction according to the Reviewer’s comments to clarify the text (P6, Lines 226-227).
Point 19. Lines 234-243: to improve clarity, I suggest using the nomenclature used in figure 6 (e.g., 2005, 2020e, 2020e+g, etc.).
Response 19: We have made correction according to the Reviewer’s comments using the nomenclature from Fig. 6 in the text (P6, Lines 256-260).
Point 20. Line 317: there is an extra “)”.
Response20: We erased the extra ")".
Point 21. The acronym “SNPs” is sometimes presented as “SNP’s” (line 392) and “SNPs” (line 437). It should be “SNPs”.
Response 21: We changed all acronyms to “SNPs”.
Point 22. Lines 393-395: the sentence “Although the differences...” is unclear.
Response 22: We modified the sentence in the text to clarify the idea (Page 11, Lines 419-427)
Point 23. Lines 410-413: this study did not assess the effect of land use change and rock climbing. I suggest clearly distinguishing the findings of this study from those of previous ones.
Response 23: We modified the sentence in the text according to the Reviewer’s comments to clarify the idea (Page 11, Lines 439-443).
Point 24. Line 416: the word “high” should not be capitalized.
Response 24: We changed “High” by “incipient” (P12, Line 445).
Point 25. Line 441: it should be “DAPC”.
Response 25: Also “DACP” by “DAPC” was changed (P12, Line 469).
Point 26. Lines 440-444: I found it difficult to understand this paragraph. What does “grouping by the similarity between analyses” mean?
Response 26: We have rewritten the redaction according to the reviewer’s comments to clarify the text (Page 12, Line 471-477)
Point 27. Lines: 476-479: I suggest using a more impersonal language when stating the outlook of the study.
Response 27: We modified the sentence in the text according to the Reviewer’s Points to clarify the idea (Page 13, Lines 511-513).

Reviewer 2 Report
Peer-Review of Change conservation status of Pachyphytum caesium (Crassulaceae), a threatened species from central Mexico based on genetic studies by Martínez-León et al.
Thank you for the opportunity to review this paper about a very interesting plant species in danger of extinction due to land-use change(s), which needs to be included in the introduction more generally, along with climate change issues affecting the focal geographic area.
Abstract: Suggest leaving out specific GD results, but present general inferences from the entirety of the work and their impact on the authors’ position on this policy-driven paper.
Introduction:
This introduction is strongly policy-driven, specific to the nation’s MER. This is an important paper for this and other endemic plant species. An effort to make the case for inclusion of genetic data, and the inferences derived, can be elaborated more or made stronger. Overall, a fairly clear introduction with only minor word choice, sentence structure issues. Some biogeography concepts probably should be considered for inclusion. Assumptions regarding the selected methodology and expectations of the genetic/scientific results should be present in the last paragraph, as well.
See works on pop gen of rare and/or invasive species by JL Hamrick, Baker & Stebbins, and other pop gen papers, for examples on presenting the data herein in a more clear and concise manner.
Minor comments:
Keywords: perhaps spell out MER, in espanol for search engines. Consider adding “biodiversity loss, threatened plant, and geographic isolation” and change “danger of extinction” to just “extinction.”
Add: Conservation genetics principles into the introduction, such as high intrinsic extinction rates based on metapopulation theory (See: Hanksi et al.)
Please define PSESP explicitly, in text.
Line 44: …can significantly strengthen [add: strategies for] population management…
Line 45: stylistic preference of the oxford comma, add comma after “…their conservation, …”
Line 48-53: Although genetic studies are essential [TOWARD the] conservation species, these results are rarely included in the IUCN [DEFINE--e.g., spell out upon first use of all acronyms, regardless of ubiquity] Red List methodology [6-9]. This omission is [WHAT? Something is missing in this sentence] even though since diverse authors suggest [awkward phrasing, unable to glean authors’ meaning] that adding the genetic dimension into the IUCN criteria would improve conservation efforts [add comma] since current parameters are [poor predictors change to “insufficient criteria”] of the [add: ecological and] evolutionary resilience of the species to global change, [add: including land-use and climatic changes] [e.g. 8-9]. Currently, the Red List is focused on external recent, current, or future threats related to sustainable habitat loss, population size, or exploitation levels [10]. [Therefore, the purpose of this study is to…]
Line 63: include land-use change prior to specifying the type of anthropogenic changes.
Materials & Methods
Figure 1, define DEM for non-expert readership in legend/caption, and in the introduction, explain/link why map is displayed with elevation (DEM), as this is a cliff-dwelling species.
Line 92: “asl” is undefined in section 2.1. Always define acronyms upon first use.
Line 92: change “seriously” to “highly or extensively” as the first word choice seems outside the realm of a scientific paper and is more appropriate for a policy report.
Line 98: The species has a restricted distribution to rocky cliffs 98
in tropical dry forest and its transition with oak forest [23-25].—Are there defined ecoregions (level 3, possibly?) that can be used here to provide additional information?
Lines 101-102: change toàLeaves form lax terminal rosettes, are grayish to grayish-purple, and obovate-oblong leaves (?), which measure from 3 to 7 cm long, 2 to 3.5 cm wide, 0.8 to 1.2 cm thick, and are slightly convex [21,23]. (Totally wrong use of semicolons in this sentence. Also suggest a multi-paneled image, i.e., Fig 2, of pictures of the plant)
Lines 104-108: Since reproductive biology is unknown, the comparison to Echeveria is fair. However, any pheological data would strengthen this paragraph, including the flowering times. The fact that it flowers indicates sexual reproduction; whether it is cleistogamous or not, which pollinator syndrome is associated with this species, etc., may be unknown, but can be stated here that additional studies of this rare plant species is necessary.
Line 113: change to 400-m (when a measurement is an adjective, it is hyphenated; whereas when it is a noun, it is not).
Section 2.3 can be rewritten to be stronger. First, dates of collection (2009-2013) should come first and include why the 4-yr time period, followed by the n per pop, and the total N, and then how the tissues were accessed utilizing a climbing strategy and geolocations recorded (was elevation recorded?). Why were the tissues stored at -20 deg C? That causes nuclear lysing and reduces gDNA. 4 deg C or -80 deg C are the more appropriate storage. How long were the tissues stored? This impacts the quality of the gDNA and ratios for SS:DS.
Section 2.4
All proprietary products should include the FULL address of the company (i.e., Qiagen). Please add throughout.
“Medium TBE” should include a percentage concentration
Use of ISSRs should be 1) defined in the introduction along with the assumptions, pros and pitfalls, and use in other, similar plant species. 2) Lines 127-137 are redundant and needs to be edited. I don’t think the multiple parenthetical 5 x 6 = 30 by 12 primers are necessary. A simple “Identification of the 12 ISSR primers utilized in this study resulted from preliminary amplification of five P. caesium individuals; these were selected for their high-polymorphic states (see: Table 1).” Create a table of the primer sets from Wolfe et al., 1998, for clarity and simplicity.
PCR specifics can be placed into supplementary information. This way, the genetic analysis methods reads more clearly.
Be very clear about this marker in both the introduction and methods that these markers are dominant and result in a presence/absence matrix. What quality control and proofing of the matrix was conducted?
2.4.1, 2.4.2, 2.4.3 can be consolidated into a single section with a paragraph for each. GenAlEx doesn’t need to be repeated and re-cited twice in such a small paragraph on simple genetic diversity assessments. Explaining “diplotypes” in a GenAlEx AMOVA is not terribly informative and only results in the partitioning of the genetic variation w/o use of F-stats.
GenAlEx is insufficient for hierarchical AMOVAs and Arlequin (Excoffier et al., ought to be utilized, which also results in pairwise Fst, where F-statistics are more informative/robust than Gst).
Most readers will understand the MSN, as it is defined in text, and simply say it was conducted in R v…no need to explain groupings. Keep the numbering of each approach to evaluate population structure, since “six” was presented in the opening sentence, but keep it more concise.
Principle components analysis, is an ordination approach to evaluate genetic variation, and is shortened to PCoA versus PCA, which is now principle coordinates analysis. There is a difference.
Which Mantel test was used?
Suggest reading other population genetic papers to clean up this section of the methods.
Line 208, why a space between 95 and %?
Line 219: “for each simulation, all variables previously calculated were obtained.” Were utilized to parameterize the post-2018 simulations? This phrase doesn’t make sense.
Line 239: remove the word “sake”
Results:
3.1 First sentence unnecessary
“The average alleles per locus was…for all six analyzed populations, [add with a range of X# of alleles for locus Y and only Z# of alleles for locus W].
Five unique alleles (do you mean private alleles?)
Again, spacing between value and the “%” symbol is awkward. Remove the unnecessary space.
In presenting genetic diversity, it’s important to report the Ho AND the He for the reader to determine how much deviation from HWE was found in the dataset. Considering this as a rare plant, the expectation is that the deviation would be pretty high, correct?
Fig 3, put vertical lines between the panels a, b, c to make reability more clear. Change all acronyms to PCoA.
Fig 5, Use Delta K or the symbol, not “DK”.
The Evanno method has issues, which defaults to K=2, where the PCoAs and the DAPCs show more than 2 groups. “The maximum value of ΔK, mean(LnProb), and mean (similarity score) were de- 321
tected at K=2 by Evanno’s method using the total individuals (106.45, -3667.8 and 0.999, 322
respectively) and simulated 50% of populations (68.33, -7700.4 and 0.990, respectively),” (the parenthetical values are uneccssary, unless individual LnPDs were utilized to separate into clusters based on a meaningful, biological threshold.
Discussion
Basic allelic and genetic diversity should be compared to Nybom’s review of dominant and co-dominant markers.
Line 397-398: is recent in Presa Cebolletas. Based on the high values of genetic diversity [as compared to other populations?] of Cebolletas, we suggest that this population functioned in the past as a genetic refuge area [do you mean refugia?], like some’s postglacial genetic refuge [???] [59]. Suggested change: this population may have functioned as a refuge population or that the Cebolletas region served as a geographic refugium since the last glacial maxima/minima [depending on what you mean here. Also, LGM is utilized quite frequently in biogeographical contextualization of population structuring from pop gen analyses.]
Line 401: remove “according to Kimura” [why add this when the paper is already cited?]
Line 409: The Pyrenees Mountain Range? Treat as proper noun.
Section 4.2:
Line 416: A High level [why is this capitalized?]
Line 423” Do you mean Alpine or montane islands at higher elevations? “form islands”
Conclusions:
Line 494: “a strong genetic structure of its populations” why make this case if K = 2? All visualized K 2-10 should be in the Supp info at least. I have my doubts that K is actually 2 clusters. It may be due to not enough dominant primers (dominant markers require more polymorphic loci than co-dom or SNPs from full genome sequencing, skimming, RADseq, or other NGS options). That would definitely strengthen the case for the authors to modify classification of this species.
Lit Cited: ALL papers should have DOIs associated with them unless they do not have one. Spacing is also odd among differing references. Please check for consistency and completeness throughout.
Author Response
Response to Reviewer #2 Comments
Point #1. Abstract: Suggest leaving out specific GD results, but present general inferences from the entirety of the work and their impact on the authors’ position on this policy-driven paper.
Response1: We reviewed and modified the redaction in the abstract section considering reviewer’s comments to this section
Point 2. Introduction:
This introduction is strongly policy-driven, specific to the nation’s MER. This is an important paper for this and other endemic plant species. An effort to make the case for inclusion of genetic data, and the inferences derived, can be elaborated more or made stronger. Overall, a fairly clear introduction with only minor word choice, sentence structure issues. Some biogeography concepts probably should be considered for inclusion. Assumptions regarding the selected methodology and expectations of the genetic/scientific results should be present in the last paragraph, as well. See works on pop gen of rare and/or invasive species by JL Hamrick, Baker & Stebbins, and other pop gen papers, for examples on presenting the data herein in a more clear and concise manner.
Response 2: We reviewed and clarified the redaction in the introduction section, based on Reviewer’s comments. However, biogeographic aspects were not incorporated since this approach was not addressed in the study.
Point 3. Keywords: perhaps spell out MER, in espanol for search engines. Consider adding “biodiversity loss, threatened plant, and geographic isolation” and change “danger of extinction” to just “extinction.”
Response 3: We agree and spelled out MER, in English for search engines. Also, we added the keywords “biodiversity loss” and “threatened plant”; also we changed “danger of extinction” to “extinction.”
Point 4. Please define PSESP explicitly, in text.
Response 4: We capitalized the first letter of every word of PSEPS and modified the sentence to clarify the PSESP concept (Page 2, Lines 71-74).
Point 5. Line 44: …can significantly strengthen [add: strategies for] population management…
Response 5: We have made correction according to the Reviewer’s comments (P2, Line 45)
Point 6. Line 45: stylistic preference of the oxford comma, add comma after “…their conservation, …”
Response 6: We added comma after... “their conservation” (P2, Line 46).
Point 7. Line 48-53: Although genetic studies are essential [TOWARD the] conservation species, these results are rarely included in the IUCN [DEFINE--e.g., spell out upon first use of all acronyms, regardless of ubiquity] Red List methodology [6-9]. This omission is [WHAT? Something is missing in this sentence] even though since diverse authors suggest [awkward phrasing, unable to glean authors’ meaning] that adding the genetic dimension into the IUCN criteria would improve conservation efforts [add comma] since current parameters are [poor predictors change to “insufficient criteria”] of the [add: ecological and] evolutionary resilience of the species to global change, [add: including land-use and climatic changes] [e.g. 8-9]. Currently, the Red List is focused on external recent, current, or future threats related to sustainable habitat loss, population size, or exploitation levels [10]. [Therefore, the purpose of this study is to…]
Response 7: We reviewed and clarified the redaction in according to the reviewer’s Points in this sentence (P2, Lines 59-64). The purpose of this study was not included in this paragraph.
Point 8. Line 63: include land-use change prior to specifying the type of anthropogenic changes.
Response 8: We included the word “land-use change” according to the Reviewer’s comments (P2, Line 75).
Point 9. Figure 1, define DEM for non-expert readership in legend/caption, and in the introduction, explain/link why map is displayed with elevation (DEM), as this is a cliff-dwelling species.
Response 9: We incorporated DEM concept in Figure 1 legend and modified the sentence in altitudinal distribution to specify an altitudinal range of the study area (Page 3, Lines 115-119).
Point 10. Line 92: “asl” is undefined in section 2.1. Always define acronyms upon first use.
Response 10: We defined the acronym of masl (P3, Line 111)
Point 11. Line 92: change “seriously” to “highly or extensively” as the first word choice seems outside the realm of a scientific paper and is more appropriate for a policy report.
Response 11: We changed “seriously” by “highly” (P3, Line 112).
Point 12. Line 98: The species has a restricted distribution to rocky cliffs in tropical dry forest and its transition with oak forest [23-25]. —Are there defined ecoregions (level 3, possibly?) that can be used here to provide additional information?
Response 12: We reviewed and clarified the redaction in this section. We inserted ecoregions in the text. (Page 3, Lines 122-131).
Point 13. Lines 101-102: change to à Leaves form lax terminal rosettes, are grayish to grayish-purple, and obovate-oblong leaves (?), which measure from 3 to 7 cm long, 2 to 3.5 cm wide, 0.8 to 1.2 cm thick, and are slightly convex [21,23]. (Totally wrong use of semicolons in this sentence. Also suggest a multi-paneled image, i.e., Fig 2, of pictures of the plant)
Response 13: We abbreviated the descriptive information, and it was incorporated as part of the legend of Figure 2.
Point 14. Lines 104-108: Since reproductive biology is unknown, the comparison to Echeveria is fair. However, any phenological data would strengthen this paragraph, including the flowering times. The fact that it flowers indicates sexual reproduction; whether it is cleistogamous or not, which pollinator syndrome is associated with this species, etc., may be unknown, but can be stated here that additional studies of this rare plant species is necessary.
Response 14: We reviewed and clarified the redaction in this section according to the reviewer’s comments (P4, Lines 132-137).
Point 15. Line 113: change to 400-m (when a measurement is an adjective, it is hyphenated; whereas when it is a noun, it is not).
Response 15: We reviewed and rewrote the biological tissue collection section, so the change was not necessary. (P4, Lines 149-154).
Point 16. First, dates of collection (2009-2013) should come first and include why the 4-yr time period, followed by the n per pop, and the total N, and then how the tissues were accessed utilizing a climbing strategy and geolocations recorded (was elevation recorded?). Why were the tissues stored at -20 deg C? That causes nuclear lysing and reduces gDNA. 4 deg C or -80 deg C are the more appropriate storage. How long were the tissues stored? This impacts the quality of the gDNA and ratios for SS:DS.
Response 16: We rewrote the biological tissue collection section according to the reviewer’s comments (P4, Lines 149-155). The storage temperature used was -70 ºC, not -20.
Point 17. All proprietary products should include the FULL address of the company (i.e., Qiagen). Please add throughout.
Response 17: We incorporated all proprietary products in the text.
Point 18. “Medium TBE” should include a percentage concentration.
Response: We incorporate the concentration percentage in the sentence (P4, Line 160).
Point 19. Use of ISSRs should be 1) defined in the introduction along with the assumptions, pros and pitfalls, and use in other, similar plant species. 2) Lines 127-137 are redundant and needs to be edited. I don’t think the multiple parenthetical 5 x 6 = 30 by 12 primers are necessary. A simple “Identification of the 12 ISSR primers utilized in this study resulted from preliminary amplification of five P. caesium individuals; these were selected for their high-polymorphic states (see: Table 1).” Create a table of the primer sets from Wolfe et al., 1998, for clarity and simplicity.
Response 19: We made corrections according to the reviewer’s comments to clarify the introduction (P2, Lines 49-58) and 2.4 method section (P5, Lines 169-172). Also, we created Table S1 (supplementary material), which describes the ISSR primers utilized.
Point 20. PCR specifics can be placed into supplementary information. This way, the genetic analysis methods read more clearly.
Response 20: We created Tables S2, S3, and S4 to describe in detail the PCR reactions and the used program.
Point 21. Be very clear about this marker in both the introduction and methods that these markers are dominant and result in a presence/absence matrix. What quality control and proofing of the matrix was conducted?
Response 21: We made corrections according to the reviewer’s comments to clarify the text in 2.4 method section (P5, Lines 169-185).
Point 22. 2.4.1, 2.4.2, 2.4.3 can be consolidated into a single section with a paragraph for each. GenAlEx doesn’t need to be repeated and re-cited twice in such a small paragraph on simple genetic diversity assessments. Explaining “diplotypes” in a GenAlEx AMOVA is not terribly informative and only results in the partitioning of the genetic variation w/o use of F-stats.
Response 22: We eliminated the repeated and re-cited software name. Also, we reviewed and modified the redaction of this section (Page 6, Lines 201-213).
We considered inadequate consolidation into a single section of the subsections 2.4.1, 2.4.2 and 2.4.3, since each subsection is independently shown in results and discussion.
Point 23. GenAlEx is insufficient for hierarchical AMOVAs and Arlequin (Excoffier et al., ought to be utilized, which also results in pairwise Fst, where F-statistics are more informative/robust than Gst).
Response 23: We calculated and reported the Fst value in the study.
Point 24. Most readers will understand the MSN, as it is defined in text, and simply say it was conducted in R v…no need to explain groupings. Keep the numbering of each approach to evaluate population structure, since “six” was presented in the opening sentence but keep it more concise.
Response 24: We rewrote this sentence to focus on the reviewer's comments. We eliminated the text related to grouping (P5, Lines 203-205):
“Consisted of grouping multilocus genotypes by genetic distances between them…… Each multilocus genotype is represented by a node and the genetic distance by the lines connecting the nodes. The nodes are expected to be joined by the minimum genetic distance between the individuals; this allowed us to register the set of nodes connected by identical genetic distances”
Point 25. Principle components analysis is an ordination approach to evaluate genetic variation, and is shortened to PCoA versus PCA, which is now principal coordinates analysis. There is a difference.
Response 25: We evaluated the Principal Components Analysis in Adegenet package, which uses PCA acronym. PCA and PCoA analyses are different in the initial data matrix. https://adegenet.r-forge.r-project.org/files/PRstats/practical-MVAintro.1.0.pdf For this reason, we used PCA as acronym
Point 26. Which Mantel test was used?
Response 26: We modified the sentence in the text according to the Reviewer’s comments to clarify the idea (P6, Lines 222-228).
Point 27. Suggest reading other population genetic papers to clean up this section of the methods.
Response 27: We made corrections according to the reviewer’s comments to clarify the text (P6, Lines 222-228)
Point 28. Line 208, why a space between 95 and %?
Response 28: We eliminated all spaces between the value and %.
Point 29. Line 239: remove the word “sake”.
Response 29: Done (P6, Line 260).
Point 30. Line 219: “for each simulation, all variables previously calculated were obtained.” Were utilized to parameterize the post-2018 simulations? This phrase doesn’t make sense.
Response30: We rewrote this sentence according to the reviewer's comments (P6, Lines 237-239).
Results:
3.1 First sentence unnecessary
Point 31. “The average alleles per locus was…for all six analyzed populations, [add with a range of X# of alleles for locus Y and only Z# of alleles for locus W].
Response 31: We made corrections according to the reviewer’s comments to clarify the text (P7, Line 269). Also, we removed the first sentence: “Twelve ISSR oligonucleotides were used for the genetic analysis from the 15 tested in the pilot evaluation”
Point 32. Five unique alleles (do you mean private alleles?)
Response 32: We changed “unique” to “private” (P7, Line 270)
Point 33. Again, spacing between value and the “%” symbol is awkward. Remove the unnecessary space.
Response 33: Done.
Point 34. In presenting genetic diversity, it’s important to report the Ho AND the He for the reader to determine how much deviation from HWE was found in the dataset. Considering this as a rare plant, the expectation is that the deviation would be pretty high, correct?
Response 34: We carefully analyze this comment, however, we cannot provide Ho since ISSR is a dominant marker, so it does not distinguish heterozygous individuals.
Point 35. Fig 3, put vertical lines between the panels a, b, c to make reability more clear. Change all acronyms to PCoA.
Response 35: We incorporated lines among panels in Figure 3. However, we evaluated the Principal Ccomponents Analysis in the adegenet package (not the Principal Coordinates Analysis, PCoA). Adegenet uses PCA acronym and for this reason, we used it.
Point 36. Fig 5, Use Delta K or the symbol, not “DK”.
Response 36: We corrected the legend according to the reviewer’s comments, and figure 5 was modified.
Point 37. The Evanno method has issues, which defaults to K=2, where the PCoAs and the DAPCs show more than 2 groups. “The maximum value of ΔK, mean (LnProb), and mean (similarity score) were de- 321 tested at K=2 by Evanno’s method using the total individuals (106.45, -3667.8 and 0.999, 322 respectively) and simulated 50% of populations (68.33, -7700.4 and 0.990, respectively),” (the parenthetical values are unnecessary, unless individual LnPDs were utilized to separate into clusters based on a meaningful, biological threshold.
Response 37: Corrections suggested by the reviewer were made (P9, P345-354). The discrepancy between the methods analyzed is recognized, which is discussed (P12, Lines 476-479).
Discussion
Point 38. Basic allelic and genetic diversity should be compared to Nybom’s review of dominant and co-dominant markers.
Response 37: We modified the paragraph considering the reviewer's comment —several codominant vs. dominant markers reviews were cited (P10, Lines 395-400, and Lines 404-405).
Point 39. Line 397-398: is recent in Presa Cebolletas. Based on the high values of genetic diversity [as compared to other populations?] of Cebolletas, we suggest that this population functioned in the past as a genetic refuge area [do you mean refugia?], like some’s postglacial genetic refuge [???] [59]. Suggested change: this population may have functioned as a refuge population or that the Cebolletas region served as a geographic refugium since the last glacial maxima/minima [depending on what you mean here. Also, LGM is utilized quite frequently in biogeographical contextualization of population structuring from pop gen analyses.]
Response 39: We made correction according to the reviewer’s comments to clarify the text (P11, Lines 419-427).
Point 40. Line 401: remove “according to Kimura” [why add this when the paper is already cited?]
Response 40: We removed the phrase “according to Kimura”
Point 41. Line 409: The Pyrenees Mountain Range? Treat as proper noun.
Response 41: We provided the name of the study region is provided, as it appears in the study (Page 11, Lines 437-438).
Section 4.2:
Point 42. Line 416: A High level [why is this capitalized?]
Response 42: We changed “High” by “incipient” (P11, L445).
Point 43. Lines 417-419: I have some difficulties to understand the meaning. Perhaps rephrase the sentence?
Response 43: We verified and edited the language in this paragraph, and the relevant corrections were made to improve it (Page 11, Lines 445-456).
Point 44. Line 423” Do you mean Alpine or montane islands at higher elevations? “form islands”
Response 44: We corrected the redaction according to the reviewer’s comments to clarify the text (Page 1, Lines 450-451)
Conclusions:
Point 45. Line 494: “a strong genetic structure of its populations” why make this case if K = 2? All visualized K 2-10 should be in the Supp info at least. I have my doubts that K is actually 2 clusters. It may be due to not enough dominant primers (dominant markers require more polymorphic loci than co-dom or SNPs from full genome sequencing, skimming, RADseq, or other NGS options). That would definitely strengthen the case for the authors to modify classification of this species.
Response 45: We modified the redaction respect to “strong genetic structure” results (P7, Lines 299-305). Thank you for your suggestion. We are redoubling our efforts to obtain resources to analyze the genetic diversity of this type of plants at the genomic level. .
Point 46. Lit Cited: ALL papers should have DOIs associated with them unless they do not have one. Spacing is also odd among differing references. Please check for consistency and completeness throughout.
Response 45: We inserted the DOI in all references, except those that did not have it. Also, the spacing was revised.

Reviewer 3 Report
The authors present an interesting conservation genetic study on an endemic plant species. The attempt is to provide knowledge to support conservation planning. Thus, the work is very relevant. Comprehensive data analysis methods were used.
More specific comments:
-Distribution: The sampling sites for P. caesium were given, but is not clear, what the total area of distribution is. Is it the whole area shown in Figure 1? Any information on the numbers of populations and their sizes? The sizes of the studied populations?
-Is there clonal propagation that would influence the genetic structure?
-Storage of plant materials at only -20 °C (and not deep-frozen or dried) can be risky for the quality of DNA once extracted, especially if the extraction is not done quite soon.
-The primer sequences should be listed in Supplementary materials, especially since there is no real reference to them given. Why a combination of two annealing temperatures (37 °C and 54 °C) was used? Also, it is surprising that the same temperatures worked for all primers.
-Since the repeatability of ISSR markers is not ideal, the analyses should be repeated. Was this done?
-Line 165: What is “genetic frequencies “? Perhaps allele frequencies?
-Results: The numbers of alleles detected per locus is incredibly high. Are the numbers really per locus of total numbers across loci? The analyses should have been repeated to confirm the results. Pictures of gels should be included as Supplementary materials to show the quality of amplifications. Preferably, the raw data should be available as Supplementary materials.
-The use of ISSR markers is not optimal, as the authors also agree and mention that the choice was due to financial limitations. This is understandable. Yet, the amount of data produced is modest, when a limited number of suboptimal markers and only six populations were included. It is not possible to draw convincing conclusions based on the genetic data.
Author Response
Response to Reviewer 3 Comments
Point 1. Distribution: The sampling sites for P. caesium were given, but is not clear, what the total area of distribution is. Is it the whole area shown in Figure 1? Any information on the numbers of populations and their sizes? The sizes of the studied populations?
Response 1: We reviewed and clarified the redaction in species description section according to the reviewer’s comments (P4, Lines 121-146). We added Figure 2.
Point 2. Is there clonal propagation that would influence the genetic structure?
Response 2: We addressed this point in discussion section (P11, Lines 412-417)
Point 3. Storage of plant materials at only -20 °C (and not deep-frozen or dried) can be risky for the quality of DNA once extracted, especially if the extraction is not done quite soon.
Response 3: We incorporated the correct temperature (P4, Line 154).
Point 4. The primer sequences should be listed in Supplementary materials, especially since there is no real reference to them given. Why a combination of two annealing temperatures (37 °C and 54 °C) was used? Also, it is surprising that the same temperatures worked for all primers.
Response 4: We created Table S1 (supplementary material), which describes the ISSR primers utilized. Also, we created Tables S2, S3, and S4 to describe the PCR reactions and the program employed.
Point 5. Since the repeatability of ISSR markers is not ideal, the analyses should be repeated. Was this done?
Response 5: We made comments to clarify the text in 2.4 method section (P5, Lines 169-171).
Point 6. Line 165: What is “genetic frequencies “? Perhaps allele frequencies?
Response 6: We changed “genetic” to “Allelic” (P5, Line 191)
Point 7. Results: The numbers of alleles detected per locus is incredibly high. Are the numbers really per locus of total numbers across loci? The analyses should have been repeated to confirm the results. Pictures of gels should be included as Supplementary materials to show the quality of amplifications. Preferably, the raw data should be available as Supplementary materials.
Response 7: We clarified this question in method section (P5, Lines 169-184). We provided a representative gel image (Figure S1).
Point 8. The use of ISSR markers is not optimal, as the authors also agree and mention that the choice was due to financial limitations. This is understandable. Yet, the amount of data produced is modest, when a limited number of suboptimal markers and only six populations were included. It is not possible to draw convincing conclusions based on the genetic data.
Response 8: Thank you for your suggestion. We hope to obtain funds to analyze this species with other and more suitable DNA-based markers. This study analyzed all known populations of the species, with a reasonable number of primers [see 14,66], that allow us to draw conclusions about the species conservation.

Reviewer 4 Report
The research entitled “Change conservation status of Pachyphytum caesium (Crassulaceae), a threatened species from central Mexico based on genetic studies” is an interesting investigation of the genetic diversity and population structure of an interesting plant endemic to central Mexico.
The results of this research improve knowledge on its conservation status and highlight the importance of incorporating phylogeography when red listing narrow endemics.
In my opinion, it is a valuable contribution but could be improved. The manuscript is well structured while English should be ameliorated. My main concern is the missing information in Materials and Methods. Some of them, especially those related to the Conservation Status assessment, are crucial for the full understanding of the paper. I also suggest to provide further information on the species and its environment and include some relevant missing reference. Please, see my specific comments below:
L 40 I suggest to rephrase the first sentence by starting with the subject
L 41: I suggest considering these two papers:
Pimm, S. L., & Joppa, L. N. (2015). How many plant species are there, where are they, and at what rate are they going extinct?. Annals of the Missouri Botanical Garden, 100(3), 170-176.
Cowie, R. H., Bouchet, P., & Fontaine, B. (2022). The Sixth Mass Extinction: fact, fiction or speculation?. Biological Reviews.
L 43: another relevant document to be considered:
Médail, F.; Baumel, A. Using phylogeography to define conservation priorities: The case of narrow endemic plants in the Med-578 iterranean Basin hotspot. Biol. Conserv. 2018, 224, 258–266, doi:10.1016/j.biocon.2018.05.028.
L 61: check double dashes
L 61 low genetic variation in plants is not always associated with low fitness or low success. This might be mentioned.
See, for instance these references:
Lammi, A., Siikamaki, P. & Mustajarvi, K. (1999) Genetic diversity, population size, and fitness in central and peripheral populations of a rare plant Lychnis viscaria. Conservation Biology, 13, 1069–1078.
Leimu, R., Mutikainen, P. I. A., Koricheva, J., & Fischer, M. (2006). How general are positive relationships between plant population size, fitness and genetic variation?. Journal of Ecology, 94(5), 942-952.
Plenk, K., Bardy, K., Höhn, M., & Kropf, M. (2019). Long-term survival and successful conservation? Low genetic diversity but no evidence for reduced reproductive success at the north-westernmost range edge of Poa badensis (Poaceae) in Central Europe. Biodiversity and conservation, 28(5), 1245-1265.
L 69: I suggest rephrasing as following: is a cliff-dwelling species endemic to the dry...
L99-108: this part can be reduced while more inherent information can be provided , for instance, about its distribution, demography, reproduction, dispersion, and possible human uses or the absence of such critical information. Also regarding the environment, the presence of barriers, such as mountains between population should be here mentioned
L 185: functions
L 191: the version of R can be specified only once by mentioning that all analyses were done with the same version
L221: I suggest entitling the subsection “Conservation status”
L 223-224: one between criteria and method can be omitted
L 232: the conservation status assessment is a crucial point for your conclusions. Considering the globally uncommon criteria used, further specific on how these scored were calculated might be provided in the Supplementary material
Figure 5: genetic structure might be coupled with the spatial distribution of population and, if present, ecoregions or similar. There are many examples but the first that come into my mind is the fig. 1 of this paper
Gentili, R., Fenu, G., Mattana, E., Citterio, S., De Mattia, F., & Bacchetta, G. (2015). Conservation genetics of two island endemic R ibes spp.(G rossulariaceae) of S ardinia: survival or extinction?. Plant Biology, 17(5), 1085-1094.
L 348 see comment above
L 372: unclear the positioning of references and use of the comma
L 399: unclear use of Saxon genitive
L 420 these barriers should be somehow showed in a map (see comment in Fig. 5)
L 481: revise “species at national risk”
L 488: the genitic information might guide such collections. See for instance
Caujapé-Castells, J.; Pedrola-Monfort, J. Designing ex-situ conservation strategies through the assessment of neutral genetic 759 markers: Application to the endangered Androcymbium gramineum. Conserv. Genet. 2004, 5, 131–144, 760 doi:10.1023/b:coge.0000029997.59502.88.
L 499: the authors can mention that genetic data can support future conservation planning
REFERENCES: the doi is missing in some paper; the year is non-uniformly positioned (see fo instance Lanes et al)
Author Response
Response to Reviewer 4 Comments
Point 1. My main concern is the missing information in Materials and Methods. Some of them, especially those related to the Conservation Status assessment, are crucial for the full understanding of the paper. I also suggest providing further information on the species and its environment and include some relevant missing reference. Please, see my specific comments below:
Response 1: We agree with the reviewer's comments, we reviewed and clarified the redaction in the species description section (Lines 121-146). We also provided additional information of the species and its environment (P4, Lines 106-119).
Point 2. L 40 I suggest rephrasing the first sentence by starting with the subject
Response 2: We modified the redaction in the text according to the reviewer’s comments (P1-2, Lines 41-43).
Point 3. L 41: I suggest considering these two papers: Pimm, S. L., & Joppa, L. N. (2015). How many plant species are there, where are they, and at what rate are they going extinct? Annals of the Missouri Botanical Garden, 100(3), 170-176.
Cowie, R. H., Bouchet, P., & Fontaine, B. (2022). The Sixth Mass Extinction: fact, fiction, or speculation? Biological Reviews.
L 43: another relevant document to be considered:
Médail, F.; Baumel, A. Using phylogeography to define conservation priorities: The case of narrow endemic plants in the Med-578 iterranean Basin hotspot. Biol. Conserv. 2018, 224, 258–266, doi:10.1016/j.biocon.2018.05.028.
Response 3: Thank you for your suggestion. The information from the papers of Pimm et al. 2015 and Medail et al. 2018 was incorporated into the text.
Point 4. L 61: check double dashes
Response: We removed all double dashes.
Point 5. Line 165: What is “genetic frequencies “? Perhaps allele frequencies?
Response 5: We changed “genetic” by “Allelicic” (P5, Line 191)
Point 6. L 61 low genetic variation in plants is not always associated with low fitness or low success. This might be mentioned.
See, for instance these references:
Lammi, A., Siikamaki, P. & Mustajarvi, K. (1999) Genetic diversity, population size, and fitness in central and peripheral populations of a rare plant Lychnis viscaria. Conservation Biology, 13, 1069–1078.
Leimu, R., Mutikainen, P. I. A., Koricheva, J., & Fischer, M. (2006). How general are positive relationships between plant population size, fitness and genetic variation? Journal of Ecology, 94(5), 942-952.
Plenk, K., Bardy, K., Höhn, M., & Kropf, M. (2019). Long-term survival and successful conservation? Low genetic diversity but no evidence for reduced reproductive success at the north-westernmost range edge of Poa badensis (Poaceae) in Central Europe. Biodiversity and conservation, 28(5), 1245-1265.
Response 6: We did not include this information in the introduction because it was not part of the study, and no information is available for this species. However, it was mentioned in the discussion section (P11, Lines 408-412). We considered the references of Lammi et al. 1999 and Plenk et al. 2019 in the text.
Point 7. L 69: I suggest rephrasing as following: is a cliff-dwelling species endemic to the dry...
Response 7: We modified the redaction in the text according to the reviewer’s comments (P2, Line 81).
Point 8. L99-108: this part can be reduced while more inherent information can be provided, for instance, about its distribution, demography, reproduction, dispersion, and possible human uses or the absence of such critical information. Also, regarding the environment, the presence of barriers, such as mountains between population should be here mentioned
Response 8: According to the reviewer's comments, we reviewed and clarified the redaction in the species description section. We provided additional information of the species and its biology (P3-4, Lines 122-137).
Point 9. L221: I suggest entitling the subsection “Conservation status”
Response 9: We entitled the subsection in method and results section according to the reviewer’s comments (L241 and L369).
Point 10. L 191: the version of R can be specified only once by mentioning that all analyses were done with the same version.
Response 10: We modified the redaction in the methods section according to the reviewer’s comments (P5, Lines 205-212).
Point 11. L 223-224: one between criteria and method can be omitted.
Response 11: We omitted “criteria” in the phrase (P6, L243).
Point 12. L 232: the conservation status assessment is a crucial comment for your conclusions. Considering the globally uncommon criteria used, further specific on how these scored were calculated might be provided in the Supplementary material.
Response 11: We corrected the text according to the Reviewer’s comments to clarify the text (P6, Lines 251). MER analysis scores are available in [30]
Point 13. Figure 5: genetic structure might be coupled with the spatial distribution of population and, if present, ecoregions or similar. There are many examples but the first that come into my mind is the fig. 1 of this paper
Gentili, R., Fenu, G., Mattana, E., Citterio, S., De Mattia, F., & Bacchetta, G. (2015). Conservation genetics of two island endemic R. ibes spp.(G rossulariaceae) of S ardinia: survival or extinction?. Plant Biology, 17(5), 1085-1094.
Response 13: We made corrections according to the reviewer’s comments and inserted in Figure 1 the membership proportion of individuals inferred by STRUCTURE software. Also, we cited Gentili et al. 2015 in the text (P12, Lines 476-478).
Point 14. L 372: unclear the positioning of references and use of the comma.
Response 14: We made the correction according to the Reviewer’s comments to clarify the text (Page 10, Lines 395-400).
Point 15. L 399: unclear use of Saxon genitive
Response 15: We incorporated the sentence "of temperate plant species" to respond to the reviewer, because there are various species studied from temperate forests (P11, Line 427).
Point 16. L 420 these barriers should be somehow showed in a map (see comment in Fig. 5).
Response 16: Figure 1 provides an elevation model (MDE), which gives an idea of the geographic landscape. We agree that an analysis of geographic modeling will provide more information. We are currently obtaining data on each cliff's physical characteristics and climatic variables to elaborate a spatial distribution model.
Point 17. L 481: revise “species at national risk”
Response 17: We consider that the comment does not apply since this study precisely analyzed this point. In such a case, it could be a reassessment of the level of risk with new markers.
Point 18. L 488: the genetic information might guide such collections. See for instance.
Caujapé-Castells, J.; Pedrola-Monfort, J. Designing ex-situ conservation strategies through the assessment of neutral genetic 759 markers: Application to the endangered Androcymbium gramineum. Conserv. Genet. 2004, 5, 131–144, 760 doi:10.1023/b:coge.0000029997.59502.88.
Response 18: We made correction according to the Reviewer’s comments and wrote this phrase in the text (Page 13, Line 524).
Point 19. L 499: the authors can mention that genetic data can support future conservation planning
Response 19: We mentioned this sentence in the conclusion section (Page 13, Line 537)
Point 20. REFERENCES: the doi is missing in some paper; the year is non-uniformly positioned (see of instance Lanes et al)
Response 20: We inserted the doi in the missing references, except those that did not have it. Also, the spacing was revised.

Round 2
Reviewer 3 Report
The paper has improved significantly and the results appear reliable now.
Two specific comments:
-It is stated that the average of alleles per locus was 89.7 ± 3.9. I don't think this is the number per locus but rather the total number of loci detected across the set of ISSR primers used.
-There is still in places (at least in supplementary materials) "number of effective alleles". This should be "effective number of alleles".
-You could reconsider the title "Change conservation status...", perhaps rather "Change needed in conservation status..."
Author Response
Response to Reviewer 3 Comments
Point 1: You could reconsider the title "Change conservation status...", perhaps rather "Change needed in conservation status..."
Response 1: We propose a new title, which we consider more in line with the objective and results.
Point 2: It is stated that the average of alleles per locus was 89.7 ± 3.9. I don't think this is the number per locus but rather the total number of loci detected across the set of ISSR primers used.
Response2: The results were verified, which were analyzed again. The result shown is the average of allele number of the populations studied.
We reviewed and clarified the redaction. Changes were made according to the reviewer's comment (P7, L269).
Point 3: There is still in places (at least in supplementary materials) "number of effective alleles". This should be "effective number of alleles".
Response 3: We have corrected table S5 according to the reviewer’s comments.

Reviewer 4 Report
The authors have addressed all my previous concerns. I have no further comments.
Author Response
The reviewer did not comment